# Analysis of small and large subunit rDNA introns from several ectomycorrhizal fungi species

Li-hong Chen[1]*, Wei Yan[2], Ting Wang[1], Yu Wang[1], Jian Liu[3], Zhuo Yu[4]

**1** College of Horticulture and Plant Protection, Inner Mongolia Agricultural University, Huhhot, Inner Mongolia, China, **2** College of Forestry, Inner Mongolia Agricultural University, Huhhot, Inner Mongolia, China, **3** Ordos Institute of Technology, Ordos, Inner Mongolia, China, **4** College of Agronomy, Inner Mongolia Agricultural University, Huhhot, Inner Mongolia, China

* chenlihong29891@126.com

## Abstract

The small (18S) and large (28S) nuclear ribosomal DNA (rDNA) introns have been researched and sequenced in a variety of ectomycorrhizal fungal taxa in this study, it is found that both 18S and 28S rDNA would contain introns and display some degree variation in size, nucleotide sequences and insertion positions within the same fungi species (*Meliniomyces*). Under investigations among the tested isolates, 18S rDNA has four sites for intron insertions, 28S rDNA has two sites for intron insertions. Both 18S and 28S rDNA introns among the tested isolates belong to group I introns with a set of secondary structure elements designated P1-P10 helics and loops. We found a 12 nt nucleotide sequences `TAC CACAGGGAT` at site 2 in the 3'-end of 28S rDNA, site 2 introns just insert the upstream or the downstream of the12 nt nucleotide sequences. Afters sequence analysis of all 18S and 28S rDNA introns from tested isolates, three high conserved regions around 30 nt nucleotides (conserved 1, conserved 2, conserved 3) and identical nucleotides can be found. Conserved 1, conserved 2 and conserved 3 regions have high GC content, GC percentage is almost more than 60%. From our results, it seems that the more convenient host sites, intron sequences and secondary structures, or isolates for 18S and 28S rDNA intron insertion and deletion, the more popular they are. No matter 18S rDNA introns or 18S rDNA introns among tested isolates, complementary base pairing at the splicing sites in P1-IGS-P10 tertiary helix around 5'-end introns and exons were weak.

**Data Availability Statement:** Sequences have been uploaded to GenBank under the accession numbers listed in the Supporting Information files.

## Introduction

Mycorrhizal symbiosis is a common phenomenon in all terrestrial plant communities. One of the major types of mycorrhiza is the ectomycorrhiza, typically formed by almost all tree species in temperate forests [1]. For the ectomycorrhiza symbiosis which the fungus forms a mantle external to the plant root, the number of plant and fungal species involved is currently estimated to be ca. 6,000 and 20,000–25,000, respectively [2, 3]. The ecologically and economically most important forest trees (Pinaceae, Fagaceae, Betulaceae, Nothofagaceae,

**Funding:** This work was funded by the National Natural Science Founded of China (No. 3106112, No. 31260173).

**Competing interests:** The authors have declared that no competing interests exist.

Leptospermoideae of Myrtaceae, Dipterocarpaceae, and Amhersteae of Caesalpiniaceae, and so on) dominate woodland and forest communities in boreal, Mediterranean, and temperate forests of the Northern Hemisphere and parts of South America, seasonal savanna and rain forest habitats in Africa, India and Indo-Malay as well as temperate rain forest and seasonal woodland communities of Australia [4]. Mycorrhizal infection affects the mineral nutrition and micronutrient uptake of plants [5–7]. Based on taxonomic and ecological extrapolation, an estimated 86% of terrestrial plant species acquire mineral nutrients via mycorrhizal root symbionts [3]. For example, ectomycorrhizal fungus *Cenococcum graniforme* could produce ferricrocin, alkaline phosphatase and other hydrolyases to help hosts iron nutrient and carbohydrate utilization [1, 8]. Thus, ectomycorrhiza fungi play an important role in seedling establishment and tree growth in habitats across the globe.

Group I introns are small RNAs and are found in a wide variety of organisms (e.g. in fungi, algae and in many other unicellular eukaryotes), genes (i.e. protein, rRNA and tRNA coding genes) and genomes [9–11]. Group I introns spread effifiiciently at the DNA level into intron-less cognate sites by homing process. Group I introns are characterized by the possession of a set of conversed sequences elements designated P1 and P3-P10. P4-P6 and P3-P9 helical domains constitute the catalytic core elements and P1 and P10 helical the substrate domain that contains the 5' and 3' splice sites [12–15]. Based on both conversed nucleotide sequences and secondary structure characterics, group I introns are classified into five major groups (IA to IE) according to the presence/absence of peripheral paired elements [14, 16].

In this study, the sequnences and deduced secondary structures of 18S and 28S rDNA introns have been examined among several fungal species. We would like to know the introns insertion positions in 18S and 28S rDNA, intron sequence homology, and their secondary structure features. We are also interested in compairing 18S rDNA introns with 28S rDNA introns in the respect of their similarities and differences, trying to find their evolution origin between 18S and 28S rDNA introns.

## Materials and methods

### Fungal strains and DNA extraction

Tested strains were isolated from sclerotial bodies as well as mycorrhizae samples which were collected from Daqing Mountain (longitude 111.25˚-112.30˚, Latitude 40.35˚-40.57˚) with permission from Inner Mongolia Daqing Mountain Nature Reserve, Helan Mountain (longitude 105.40˚-105.58˚, Latitude 38.10˚-39.08˚) with permission from Helan Mountain National Nature Reserve, Daxingan Mountainn (longitude 121.30˚-121.31˚, Latitude 50.49˚-50.51˚) with permission from Genhe ecological positioning station in Daxingan Mountainn of Inner Mongolia, and Wula Mountain (longitude 108.2˚-108.5˚, Latitude 40.9˚- 40.41˚) with permission from Inner Mongolia Wula Mountain National Forest Park in Inner Mongolia of China. No specific permits were required as the research did not include the destruction of vegetation. Information regarding the used isolates is provided in Table 1. For DNA extraction, mycelial plugs from stock cultures were grown on potato-dextrose agar (PDA) plates at 24˚C for DNA extraction. Genomic DNA was extracted using a cetyltrimethyl ammonium bromide (CTAB) method [17], then stored at -20˚C.

### PCR amplification and sequencing

The 3'-end of 18S rDNA was amplified using primers NS5 (5'–GATACCGTCGTATCTTAACC-3') / NS8 (5'–TCCGCAGGTTCACCTACGGA-3') [15]. An initial denaturation at 94˚C for 5min was followed by 30 cycles of denaturation at 94˚C for 30s, annealing at 50˚C for 30s, and extension at 72˚C for 90s. There was a final extension step at 72˚C for 10min. The 3'-end

**Table 1. Isolates used in this study.**

| Isolates | Host origin | Geographical origin |
|---|---|---|
| Spop1 (*Cenococcum geophilums*) | *Populus davidiana* | Daqing Mountain, China |
| Spop2 (*Cenococcum geophilums*) | *Populus davidiana* | Daqing Mountain, China |
| Spop3 (*Cenococcum geophilums*) | *Populus davidiana* | Daqing Mountain, China |
| Spop6 (*Cenococcum geophilums*) | *Populus davidiana* | Daqing Mountain, China |
| Spopx (*Cenococcum geophilums*) | *Populus davidiana* | Daqing Mountain, China |
| Pop4 (*Cenococcum geophilums*) | *Populus davidiana* | Daqing Mountain, China |
| Pop5 (*Cenococcum geophilums*) | *Populus davidiana* | Daqing Mountain, China |
| Pop2 (*Chaetothyriales*) | *Populus davidiana* | Daqing Mountain, China |
| Pop7 (*Chaetothyriales*) | *Populus davidiana* | Daqing Mountain, China |
| Yang1 (*Cenococcum geophilums*) | *Populus davidiana* | Daqing Mountain, China |
| SHY (*Cladophialophora*) | *Populus davidiana* | Daqing Mountain, China |
| O1 (*Cenococcum geophilums*) | *Ostryopsis daidiana.* | Daqing Mountain, China |
| O2 (*Cenococcum geophilums*) | *Ostryopsis daidiana.* | Daqing Mountain, China |
| O4 (*Cenococcum geophilums*) | *Ostryopsis daidiana.* | Daqing Mountain, China |
| O5 (*Cenococcum geophilums*) | *Ostryopsis daidiana.* | Daqing Mountain, China |
| SO1 (*Cenococcum geophilums*) | *Ostryopsis daidiana.* | Daqing Mountain, China |
| SO2 (*Pezizomycotina*) | *Ostryopsis daidiana.* | Daqing Mountain, China |
| SO4 (*Cenococcum geophilums*) | *Ostryopsis daidiana.* | Daqing Mountain, China |
| SO5 (*Cenococcum geophilums*) | *Ostryopsis daidiana.* | Daqing Mountain, China |
| Picea (*Meliniomyces*) | *Picea asperata* | Daqing Mountain, China |
| Spicea (*Cenococcum geophilums*) | *Picea asperata* | Daqing Mountain, China |
| B2 (*Cladophialophora*) | *Betula platypylla* | Daqing Mountain, China |
| B3 (*Cladophialophora*) | *Betula platypylla* | Daqing Mountain, China |
| B5(*Cladophialophora*) | *Betula platypylla* | Daqing Mountain, China |
| SB1 (*Cenococcum geophilums*) | *Betula platypylla* | Daqing Mountain, China |
| SB5 (*Cenococcum geophilums*) | *Betula platypylla* | Daqing Mountain, China |
| SB6 (*Pezizomycotina*) | *Betula platypylla* | Daqing Mountain, China |
| Quercus (*Cenococcum geophilums*) | *Quercus monogolica* | Daqing Mountain, China |
| MY (*Cenococcum geophilums*) | *Pinus tabulaeformis* | Daqing Mountain, China |
| Yang2 (*Meliniomyces*) | *Populus davidiana* | Daxingan Mountain, China |
| 2010cg (*Cenococcum geophilums*) | *Betula platypylla* | Daxingan Mountain, China |
| Baihua (*Meliniomyces*) | *Betula platypylla* | Daxingan Mountain, China |
| Shanbai (*Meliniomyces*) | Unknown | Daxingan Mountain, China |
| WL (*Cenococcum geophilums*) | *Populus davidiana* | Wula Mountain, China |
| 1–1 (*Cenococcum geophilums*) | *Picea asperata* | Helan Mountain, China |
| 1–2 (*Cenococcum geophilums*) | *Picea asperata* | Helan Mountain, China |
| 1–3 (*Cenococcum geophilums*) | *Picea asperata* | Helan Mountain, China |
| YUN (*Cenococcum geophilums*) | *Picea asperata* | Helan Mountain, China |
| 2–1 (*Cenococcum geophilums*) | *Populus davidiana* | Helan Mountain, China |
| 2–2 (*Cenococcum geophilums*) | *Populus davidiana* | Helan Mountain, China |
| 2–3 (*Cenococcum geophilums*) | *Populus davidiana* | Helan Mountain, China |
| 2–4 (*Cenococcum geophilums*) | *Populus davidiana* | Helan Mountain, China |
| 2–5 (*Cenococcum geophilums*) | *Populus davidiana* | Helan Mountain, China |
| 2–6 (*Cenococcum geophilums*) | *Populus davidiana* | Helan Mountain, China |
| 2–7 (*Cenococcum geophilums*) | *Populus davidiana* | Helan Mountain, China |
| 2–8 (*Cenococcum geophilums*) | *Populus davidiana* | Helan Mountain, China |
| 2–9 (*Cenococcum geophilums*) | *Populus davidiana* | Helan Mountain, China |

*(Continued)*

**Table 1.** (Continued)

| Isolates | Host origin | Geographical origin |
|---|---|---|
| 2–10 (*Cenococcum geophilums*) | *Populus davidiana* | Helan Mountain, China |
| 2–11 (*Cenococcum geophilums*) | *Populus davidiana* | Helan Mountain, China |
| 2–12 (*Cenococcum geophilums*) | *Populus davidiana* | Helan Mountain, China |
| 2–13 (*Cenococcum geophilums*) | *Populus davidiana* | Helan Mountain, China |
| 2–14 (*Cenococcum geophilums*) | *Populus davidiana* | Helan Mountain, China |
| 2–15 (*Pezizomycotina*) | *Populus davidiana* | Helan Mountain, China |
| 2–16 (*Chaetothyriales*) | *Populus davidiana* | Helan Mountain, China |
| 2–17 (*Phialophore verrucosa*) | *Populus davidiana* | Helan Mountain, China |
| 3–1 (*Cenococcum geophilums*) | *Pinus tabulaeformis* | Helan Mountain, China |
| 3–2 (*Cenococcum geophilums*) | *Pinus tabulaeformis* | Helan Mountain, China |
| 3–3 (*Cenococcum geophilums*) | *Pinus tabulaeformis* | Helan Mountain, China |
| 3–4 (*Cenococcum geophilums*) | *Pinus tabulaeformis* | Helan Mountain, China |
| 4–1 (*Cenococcum geophilums*) | *Jumiperus communis* | Helan Mountain, China |
| CG (*Cenococcum geophilums*) | Unknown | France |
| CG5 (*Cenococcum geophilums*) | Unknown | France |
| CG54 (*Cenococcum geophilums*) | Unknown | France |
| CG417 (*Cenococcum geophilums*) | Unknown | France |
| AM51 (*Meliniomyces*) | Unknown | France |
| CGTAR (*Cenococcum geophilums*) | Unknown | Switzerland |
| CGPIL (*Cenococcum geophilums*) | Unknown | Switzerland |
| CGLESPAC (*Cenococcum geophilums*) | Unknown | Switzerland |
| 010 (*Cenococcum geophilums*) | *Pinus resinosa* Ait. | USA |
| 011 (*Cenococcum geophilums*) | *Pinus resinosa* Ait. | USA |
| 155 (*Cenococcum geophilums*) | *Quercus alba* L. | USA |
| ALB-2 (*Cenococcum geophilums*) | *Abies lasiocarpa* Nutt. | USA |
| S8-1 (*Cenococcum geophilums*) | *Picea glauca* Vess. | USA |
| HUNT-8 (*Cenococcum geophilums*) | *Picea rubrens* Sargent | USA |
| HUNT-9 (*Cenococcum geophilums*) | *Picea rubrens* Sargent | USA |
| 1-1-4 (*Cenococcum geophilums*) | *Quercus douglasii* | USA |
| 1-7-7 (*Cenococcum geophilums*) | *Quercus douglasii* | USA |
| 1-7-8 (*Cenococcum geophilums*) | *Quercus douglasii* | USA |
| 1-7-11 (*Cenococcum geophilums*) | *Quercus douglasii* | USA |
| 1-19-1 (*Cenococcum geophilums*) | *Quercus douglasii* | USA |
| 2-3-1 (*Cenococcum geophilums*) | *Quercus douglasii* | USA |
| 2-6-1 (*Cenococcum geophilums*) | *Quercus douglasii* | USA |
| 2-10-3 (*Cenococcum geophilums*) | *Quercus douglasii* | USA |
| 2-11-1 (*Cenococcum geophilums*) | *Quercus douglasii* | USA |
| 2-13-2 (*Cenococcum geophilums*) | *Quercus douglasii* | USA |
| 2-14-4 (*Cenococcum geophilums*) | *Quercus douglasii* | USA |
| 3-2-5 (*Cenococcum geophilums*) | *Quercus douglasii* | USA |
| 3-7-3 (*Cenococcum geophilums*) | *Quercus douglasii* | USA |
| 3-9-2 (*Cenococcum geophilums*) | *Quercus douglasii* | USA |
| 3-10-2 (*Cenococcum geophilums*) | *Quercus douglasii* | USA |
| 3-10-3 (*Cenococcum geophilums*) | *Quercus douglasii* | USA |
| 3-11-1 (*Cenococcum geophilums*) | *Quercus douglasii* | USA |
| 3-18-1 (*Cenococcum geophilums*) | *Quercus douglasii* | USA |
| 1-5-4 (*Cenococcum geophilums*) | *Quercus douglasii* | USA |

(*Continued*)

**Table 1.** (Continued)

| Isolates | Host origin | Geographical origin |
|---|---|---|
| 3-4-II (*Cenococcum geophilums*) | *Quercus douglasii* | USA |
| I-2 (*Cenococcum geophilums*) | *Quercus douglasii* | USA |
| I-3 (*Cenococcum geophilums*) | *Quercus douglasii* | USA |
| BTREE1 (*Cenococcum geophilums*) | *Quercus douglasii* | USA |

of 28S rDNA was amplified using primers Vdahl4 (5′-CGGGCTTGGCAGAATCAG-3′) / Vdahl2 (5′-GCGACGTCGCTATGAACG-3′) [18]. An initial denaturation at 94˚C for 1min was followed by 30 cycles of denaturation at 94˚C for 30s, annealing at 47˚C for 30s, and extension at 72˚C for 90s. There was a final extension step at 72˚C for 10min. 18S rDNA-ITS-28S rDNA region was amplified using primers ITS1 (5′-TCCGTAGGTGAACCTGCGG-3′) / ITS4 (5′-TCCT CCGCTTATTGATATGC-3′) [19]. An initial denaturation at 94˚C for 1min was followed by 30 cycles of denaturation at 94˚C for 30s, annealing at 50˚C for 30s, and extension at 72˚C for 120s. There was a final extension step at 72˚C for 10min. The products were electrophoresed in a 1% (w/v) agarose gel to check the efficiency of amplification. The purified amplicons were sequenced by Shanghai Sangon Biotechnology Co., Ltd, Shanghai Invitrogen Biotechnology Co., Ltd, Beijing Tsingke Biotechnology Co., Ltd., China. The sequences were aligned by sequence analysis software DNAMAN, Lynnon Corporation.

### Intron secondary structure modeling

Secondary structure models were predicted following the conventions for group I introns defined by Burke et al. and according to the models proposed by Cech and Michel and Westhof [12–14]. The P1-P9 stem-loop elements were individually identified by comparison with available group I intron sequences from the Comparative RNA web site (CRW at http://www.rna.icmb.utexas.edu/) and then folded using the mfold web server at http://www.bioinfo.rpi.edu/applications/mfold/old/rna/form1.cgi [20, 21]. The RNA secondary structures were calculated and drawn using RNAstructure version 4.6 [22].

## Results

### Positions and structure analysis of 18S rDNA introns

The 18S rDNA 3'-end of tested isolates (AM51, Baihua, Shanbai, Picea, Yang2, Pop7, SB6, SO2, B2, B3, B5, 2–15, 2–16, 2–17, SHY) was PCR amplified by primers NS5 / NS8, 18S rDNA-ITS-28S rDNA region isolates (SB6, SO2) was PCR amplified by primers ITS1 / ITS4, and then sequenced. After sequencing it was found that the isolates AM51, Baihua, Picea, Shanbai, Yang2, Pop7, SO2, SB6 possessed the introns, while the isolates 2–15, 2–16, 2–17, B2, B3, B5, SHY did not contain introns in 18S rDNA 3'-end. We found 18S rDNA of the tested isolates has four sites for intron insertions, the introns (Picea-I1, Pop7-I) insert at the same site in 18S rDNA sequence (site 1), the intron (AM51-I) insert at site 2, the introns (Picea-I2, Yang2-I, Baihua-I, Shanbai-I, Spop1-I, O5-I) insert at site 3, the introns (SB6-I, SO2-I) insert at site 4. Isolate Picea has two different type introns (Picea-I1 and Picea-I2) at the 3'-end of 18S rDNA, distributing at site 1 and site 3. The 18S rDNA full length of isolates Picea, Shanbai, AM51, Spop1, O5, CG54 were sequenced, there was no introns found at the 5'-end of 18S rDNA. The intron distribution in 18S rDNA of tested isolates in this study was showed in Fig 1, the exon sequences flanking introns were showed in Fig 2.

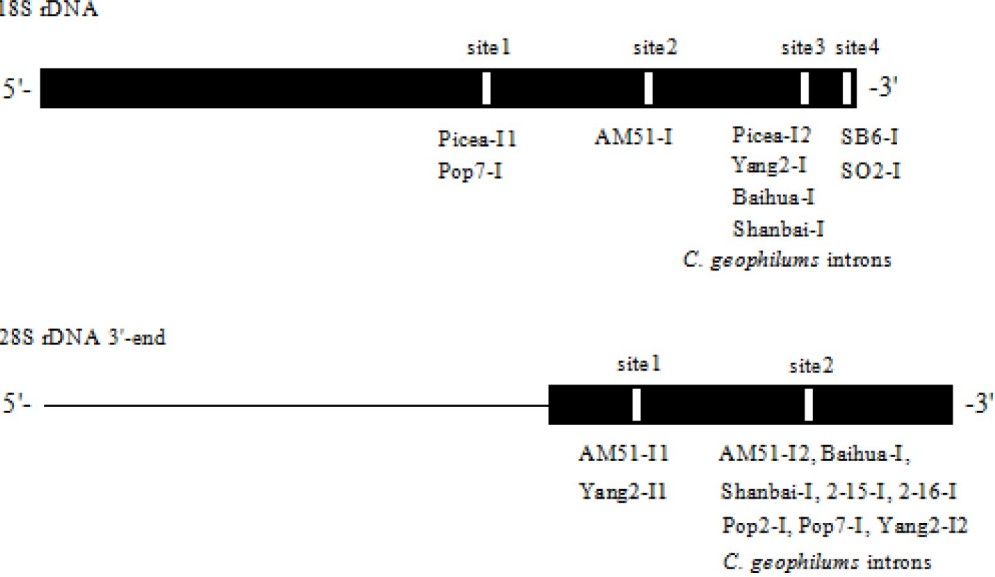

**Fig 1. The positions of intron insertion in 18S and 28S rDNA of tested isolates.**

Fig 3 showed that the deduced secondary structures of 18S rDNA introns (AM51-I, SB6-I, Pop7-I, Picea-I1, Picea-I2, 1-1-I) from tested isolates had the same features known to be conserved among group-I introns: the last exon base U and the last intron base G; the pairing regions P1-P10; the consensus elements P, Q, R and S within the core region; the internal guide sequences (IGS) proposed to help align the exons for splicing [23–29]. Beside these common structures of group-I introns above, the 18S rDNA introns (Picea-I1, Picea-I2, Pop7-I, SB6-I, 1-1-I, Spicea-I) have an extensive P5 region (P5, P5a, P5b, P5c and P5d), the 18S rDNA introns (Picea-I1, Picea-I2, Pop-I, AM51-I, 1-1-I, Spicea-I) have two extra stems on the 3' side of P9 (P9.1 and P9.2) from this study and we reported previously [30, 31]. The 18S rDNA intron (Picea-I1, Picea-I2, Pop7-I, AM51-I, 1-1-I, Spicea-I) possess an A-rich bulge, however, we did not find an typical A-rich bulge around P5 pairing region in the secondary structures of 18S rDNA intron SB6-I. The sequences of Picea-I2, Yang2-I, Baihua-I, Shanbai-I exhibited 94.7% identity, they have the same secondary structure. The sequences of SB6-I and SO2-I

## Exon sequences flanking introns in18S rDNA

5'-GGAAGGGCACCACCAGGAGT---intron---GGAGCCTGCGGCTTAATTTGAC-3', site 1
5'-GGAAGTTTGAGGCAATAACAGG---intron---CTGTGATGCCCTTAGATGTT-3', site 2
5'-AGTAAAAGTCGTAACAAGGT---intron---TCCGTAGGTGAACCTGCGGA-3', site 3
5'-TTCGTAGTGAACCTGCGGAGGGATCAT---intron---TAGACCCCTCCGGGGGTTTA-3', site 4

## Exon sequences flanking introns in 28S rDNA

5'-TTGGCAGAATCAGCGGGGAAAGAAGACCCT---intron---GTTGAGCTTGACTC TAGTTTGACA-3', site 1
5'-TAGAGGTGCCAGAAAAGT<u>TACCACAGGGAT</u>---intron---AACTGGCTTGTGGCAGCCAAGCGT-3', site 2 (Shanbai-I)
5'-TAGAGGTGCCAGAAAAGT---intron---<u>TACCACAGGGATA</u>ACTGGCTTGTGGCAGCCAAGCGT-3', site 2 (Pop7-I)

**Fig 2. The exon sequences flanking introns in 18S and 28S rDNA of tested isolates.** Exon sequences flanking introns in 28S rDNA, site 1, 5'-end sequences from *Pezizomycotina* 28S rDNA in GenBank, 3'-end sequences from isolate AM51 this study.

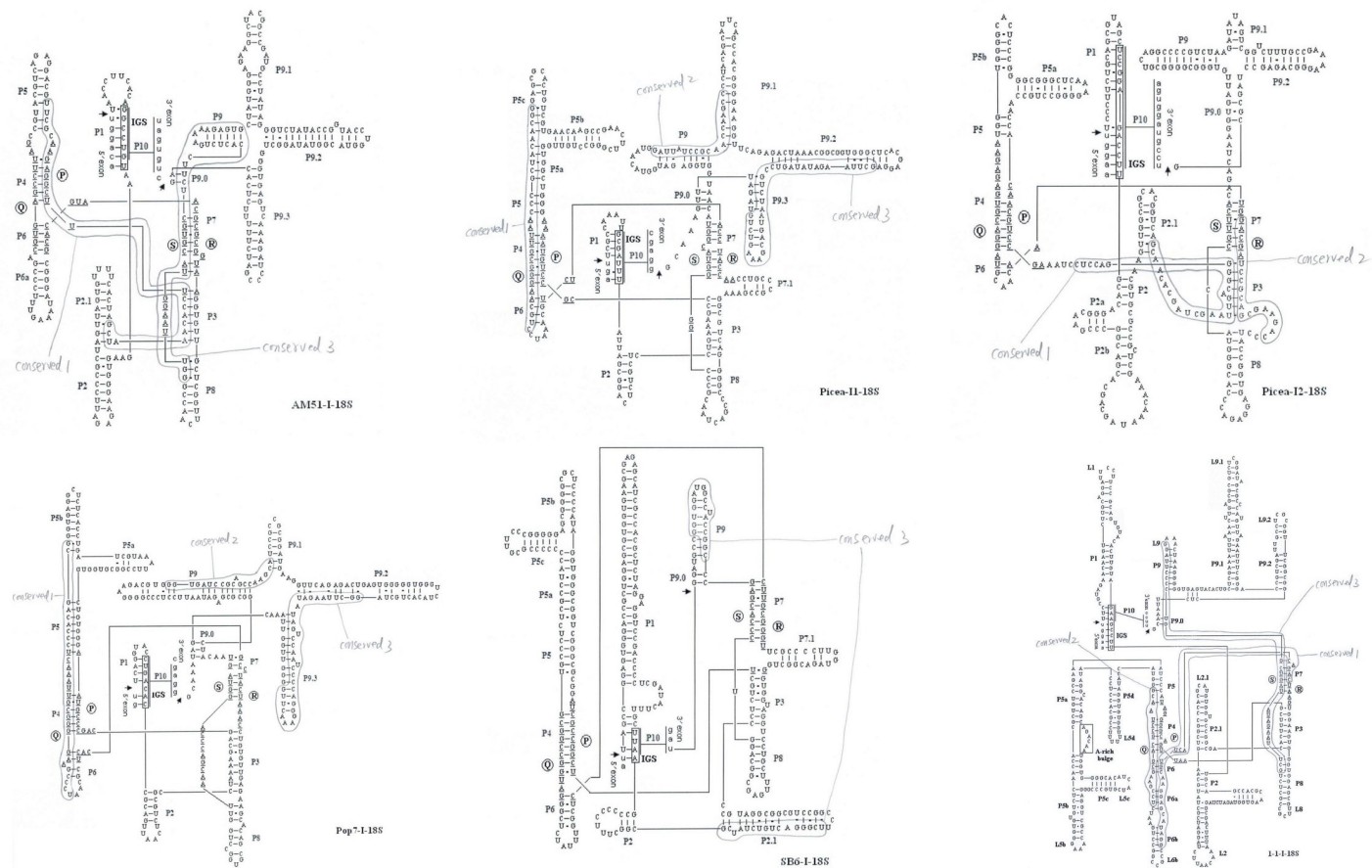

**Fig 3. Secondary structures of 18S rDNA introns AM51-I, SB6-I, Pop7-I, Picea-I1, Picea-I2, 1-1-I.** The nucleotides of the 18S rDNA intron are indicated in capital letters, while the flanking exons are in lower case letters. Arrows denote the 5' and 3' splice sites. Nucleotides within the conserved core element P, Q, R and S regions are underlined. The IGS region and A-rich bulge are indicted by boxes surrounding the sequences. Conerved 1, conserved 2, and conserved 3 regions are rounded by light line.

exhibited 98.8% identity, they have the same secondary structure. Picea-I1 and Pop-I have quite low sequence identity (61%), but still have quite similar secondary structures.

## Positions and structure analysis of 28S rDNA introns

The 28S rDNA 3'-end of tested isolates (Spop1, Spop2, Spop3, Pop4, Pop5, Spop6, Spopx, Pop2, Pop7, O1, O2, O4, O5, SO1, SO2, SO4, SO5, SB1, SB2, SB5, SB6, 1–2, 2–1, 2–2, 2–4, 2–5, 2–6, 2–7, 2–8, 2–9, 2–10, 2–12, 2–14, 2–15, 2–16, 2–17, 3–1, 3–3, 3–4, 4–1, WL, 2010cg, MY, AM51, Baihua, Shanbai, Yang2, B2, B3, B5, CG5, CG417, CG54) was amplified by PCR and sequenced. After sequencing it was found that the isolates Spop1, Spop3, Pop4, Spop6, Spopx, O1, O2, O4, O5, SO1, SO5, SB1, SB5, 2–2, 2–5, 2–6, 2–7, 2–8, 2–12, 2–15, 2–16, 3–1, 3–4, WL, 2010cg, CG5, CG417, CG54, AM51, Yang2, Baihua, Shanbai, Pop2, Pop7 possessed introns, the isolates Pop5, Spop2, MY, SO2, SO4, SB2, SB6, 1–2, 2–1, 2–4, 2–9, 2–10, 2–14, 2–17, 3–3, 4–1, B2, B3, B5 did not have introns. 28S rDNA 3'-end has two sites for intron insertions (Fig 1). Except isolates AM51 and Yang2 have two types introns (AM51-I1, AM51-I2, Yang2-I1, Yang2-I2) and insert at site 1 and site 2, the other introns (Shanbai-I, Baihua-I, Pop2-I, Pop7-I, 2-15-I, 2-16-I, and all *Cenococcum geophilums* introns) insert at site 2. We found a 12 nt nucleotide sequences TACCACAGGGAT at site 2 in the 3'-end of 28S rDNA. Introns

AM51-I2, Baihua-I, Picea-I, Shanbai-I, 2-15-I, 2-16-I, and all tested *Cenococcum geophilums* introns just insert in the downstream of the12 nt nucleotide sequences, while introns Pop2-I, Pop7-I just insert in the upstream of the 12 nt nucleotide sequences (Fig 2). The intron distribution in 28S rDNA of tested isolates in this study was showed in Fig 1, the exon sequences flanking introns were showed in Fig 2. Intron distribution compairson between18S rDNA and 28S rDNA were listed in Table 2. Some isolates have both 18S and 28S rDNA introns, some isolates have one of 18S or 28S rDNA introns, some isolates have neither 18S or 28S rDNA introns. Among tested isolates, AM51, Picea, Yang2, Shanbai, Baihua, belong to *Meliniomyces* spesice, both 18S and 28S rDNA introns display some degree variation in size, nucleotide sequences and insertion positions. While all tested *Cenococcum geophilums* 18S introns insert at site 3 and 28S introns insert at site 2, both sequences display high homology, respectively.

Fig 4 showed that the deduced secondary structures of 28S rDNA introns (AM51-I1, AM51-I2, Shanbai-I, Pop7-I, 2-15-I, 2-16-I, O1-I, SO5-I) from the tested isolates had the same features known to be conserved among group-I introns: the last exon base U and the last intron base G; the pairing regions P1-P10; the consensus elements P, Q, R and S within the core region; the internal guide sequences (IGS) necessary for alignment of the two exons for splicing; the same insertion positions (site 2) compared with other group-I introns. Beside these common structures of group-I introns above, all tested 28S rDNA introns have an A-rich bulge around P5 pairing region, an more or less extensive P5 region, and extra stems on the 3' side of P9 (P9.1, P9.2, P9.3).

Sequence analysis of 28S rDNA site 2 introns (AM51-I2, Yang2-I2, Picea-I, Shanbai-I, Baihua-I, Pop2-I, Pop7-I, 2-15-I, 2-16-I, and all *Cenococcum geophilums* introns) from tested isolates, it was found three high conserved regions around 30 nt nucleotides (conserved 1, conserved 2, conserved 3), and identical nucleotides can be found in the three conserved regions (Fig 5). Conserved 1, conserved 2 and conserved 3 regions have high GC content, GC percentage is almost more than 60%, that implied conserved 1, conserved 2, conserved 3 regions take part in complementary base pairing which maybe more firm. Sequence analysis of

**Table 2. Intron distribution patterns of 18S and 28S rDNA in tested isolates.**

| Isolate | 18S Intron | 28S Intron | Isolate | 18S Intron | 28S Intron | Isolate | 18S Intron | 28S Intron | Isolate | 18S Intron | 28S Intron |
|---|---|---|---|---|---|---|---|---|---|---|---|
| O1 | – | + | Pop5 | + | + | YUN | + | + | 2–13 | + | + |
| O2 | – | + | Yang1 | + | UN | 2–1 | – | – | 2–14 | – | – |
| O4 | + | + | Quercus | – | UN | 2–2 | – | + | 3–1 | + | + |
| O5 | + | + | 2010cg | + | + | 2–3 | – | + | 3–2 | + | UN |
| SO1 | + | + | SB1 | + | + | 2–4 | + | – | 3–3 | + | – |
| SO4 | – | – | SB2 | – | – | 2–5 | – | + | 3–4 | – | + |
| SO5 | – | + | SB5 | + | + | 2–6 | – | + | 4–1 | + | – |
| Spop1 | + | + | Spicea | + | + | 2–7 | – | + | CG5 | + | + |
| Spop2 | + | – | MY | + | – | 2–8 | – | + | CG417 | + | + |
| Spop3 | + | + | WL | – | + | 2–9 | + | – | CG5 | + | + |
| Spop6 | + | + | 1–1 | + | UN | 2–10 | – | – | CG | + | UN |
| Spopx | + | + | 1–2 | + | – | 2–11 | – | + | 2–15 | – | + |
| Pop4 | + | – | 1–3 | + | UN | 2–12 | – | + | 2–16 | – | + |
| 2–17 | – | – | Picea | + | UN | SB6 | + | – | SO2 | + | – |
| AM51 | + | + | Pop7 | + | + | Pop2 | UN | + | Yang2 | + | + |
| Shanbai | + | + | Baihua | + | + | B3 | – | – | B5 | – | – |
| SHY | – | UN | B2 | – | – | | | | | | |

"+": presence of intron; "–": absence of intron; "UN": unknown

the three high conserved regions combining with deduced intron RNA secondary structures, three high conserved regions maybe participate in forming P3, P7, P4, helices- core region (the consensus elements P, Q, R and S within the core region), or important for maintaining core region structure, or splicing founction. Conserved 1 region distributes around P3 and P4 helices, and can pull P3 and P4 helices together. Conserved 2 region distributes around P4, P6, P7 helices, that maybe make P Q consensus elements in P4 helix more stable (conserved 2 region can pair with conserved 1 region in many introns, for example AM51-I2, Shanbai-I, 2-15-I, and all tested *Cenococcum geophilums* introns.), or can pull P6 and P7 helices together (conserved 2 region distributes around P6 and P7 helices in introns Pop7-I and Pop2-I). Conserved 2 region in intron 2-16-I can be found in P9 helix unpairing region, in which small ORF can be found. Conserved 2 region did not be found in intron SO5-I. Conserved 3 region distributes around P7, P8, P9, maybe important for strengthening core region secondary structure, or important for forming loop L8, L9, L9.1, L9.2, L9.3 (Fig 4). According to their distributions in introns, there are three conditions: (1) Conserved 1, conserved 2, and conserved 3 regions all maybe pull the consensus elements P, Q, R and S together to make the core region of secondary structure more stable and form loop L9 in tested introns AM51-I2, Yang2-I2, Picea-I, 2-15-I, Shanbai-I, Baihua-I, and all *Cenococcum geophilums* introns; (2) Conserved 1 and conserved 2 regions maybe pull the elements P, Q, R and S together, or make the core region more stable in tested introns Pop2-I, Pop7-I, conserved 3 region maybe important for P9 helice to form loop L9.1a; (3) Introns 2-16-I and SO5-I, only conserved 1 maybe pull the elements P, Q, R and S together, conserved 2 and conserved 3 maybe important for P9 helix to form loop L9 and L9.3. (1) type has majority tested introns, (1) type introns maybe more stable, suitable or highly efficient for intron insertion and deletion. Comparing tested intron sequences,

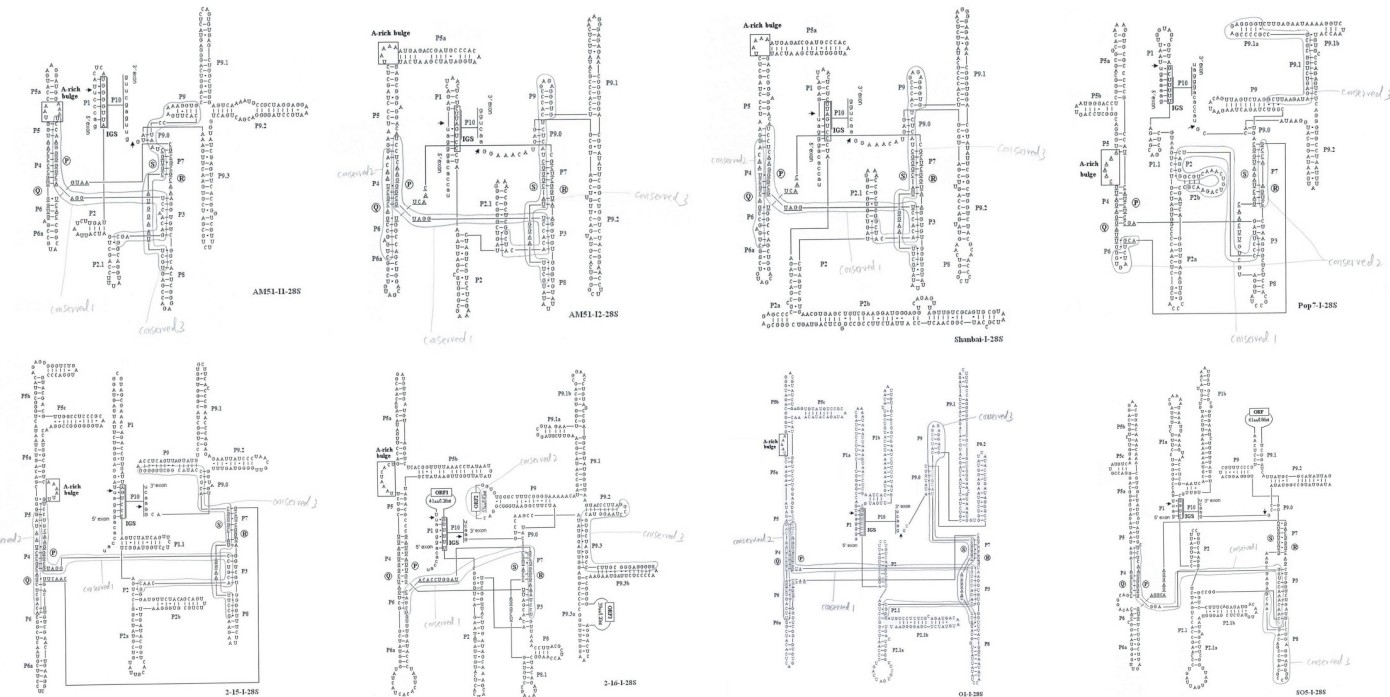

**Fig 4. Secondary structures of 28S rDNA introns AM51-I1, AM51-I2, Shanbai-I, Pop7-I, 2-15-I, 2-16-I, O1-I, O1-I, SO5-I.** The nucleotides of the 18S rDNA intron are indicated in capital letters, while the flanking exons are in lower case letters. Arrows denote the 5' and 3' splice sites. Nucleotides within the conserved core element P, Q, R and S regions are underlined. The IGS region and A-rich bulge are indicted by boxes surrounding the sequences. Conerved 1, conserved 2, and conserved 3 regions are rounded by light line.

conserved 1 region is more conservative than conserved 2 and conserved 3 regions. Conserved 3 region seems more conservative than conserved 2 region. Conserved 1 region seems more important for intron core region structure maintaining. Conserved 1, conserved 2 and conserved 3 regions in introns 2-16-I and SO5-I, containing long unpairing nucleotide sequence with small HEG ORFs, overall are less conservative than introns without HEG ORFs. The introns containing HEGs can be spliced by homing endonucleases, and endonuclease-mediated intron homing is an effifficient process. Homing is initiated by an intron-encoded homing endonuclease that recognizes and generates a double-stranded DNA break close to the site of intron insertion [32–40]. Because introns containing HEGs can code themself endonucleases to splice introns, probably they did not need conserved sequences too much, or dependent on conserved sequences completely. This maybe the reason why sequences of introns containing HEGs are less conservative than introns without HEGs.

Sequence analysis of 28S rDNA site 1 introns (AM51-I1 and Yang2-I1) from isolates AM51 and Yang2, conserved 1 and conserved 3 regions still can be found. Sequence analysis of conserved 1, 3 regions combining with intron secondary structures, conserved 1 region distributes around P3 and P4 helices and can pull them together, conserved 3 region distributes around P7, P8, P9, maybe important for strengthening core region secondary structure, or important for forming loop L9 (Fig 5). Conserved 2 region did not find in introns AM51-I1 and Yang2-I1.

We would try to find out whether the 28S intron conserved 1, 2, 3 regions exist in 18S rDNA introns or not, interestingly the trace of 28S intron conserved 1, 2, 3 regions can be found in 18S rDNA introns (Figs 5 and 6). Conserved 1, conserved 2 and conserved 3 can be found in all *Cenococcum geophilums* 18S rDNA introns listed in Table 1 (site 3), differently just conserved 2 located in the upstream of conserved 1, but conserved 2 still can pair with conserved 1 (Fig 6). *Cenococcum geophilums* is an ecologically important ectomycorrhizal fungus with a global distribution and a broad host range [41], if there is a reason because its 18S and 28S rDNA intron sequences and secondary structures are easy for insertion and deletion? Conserved 1, conserved 2 and conserved 3 can be found in 18S rDNA introns Picea-I1 and Pop7-I (site 1). Conserved 1 and conserved 2 can be found in 18S rDNA introns Picea-I2, Yang2-I, Baihua-I, Shanbai-I (site 3). Conserved 1 and conserved 3 can be found in 18S rDNA intron AM51-I (site 2). Only conserved 3 can be found in 18S rDNA intron SB6-I (site 4), but was divided into two part, 5'-end located in P2.1 helix, 3'-end located in helix P9 and loop L9 (Fig 3).

## Discussion

Intron 2-16-I and SO5-I, beside pairing regions P1-P10, they have long unpairing regions, try to find open reading frame and seem they contain small ORFs, maybe they belong to HEG-associated group I introns (Fig 4). Goddard and Burt (1999) published a model of intron life-cycle and homing that involved intron cyclical gain and loss. Full-length HEG maybe need for invading, once the intron becomes fixed, the HEG no longer need, therefore it will accumulate mutations and become non-founctional or lost HEG [42]. From this evoluation point of view, the introns without HEG genes maybe advanced, the introns containing HEG genes maybe old. We found conserved 1, 2, 3 regions from introns 2-16-I and SO5-I with HEG are less conservative than as the introns without HEG did. Introns containing HEG are very rare among 18S rDNA and 28S rDNA, we only found three introns containing HEG (SB5-I from 18S rDNA, SO5-I and 2-16-I from 28S rDNA) from our all tested 18S rDNA and 28S rDNA sequences. The HEG gene no longer need, will be gradually deleted, 2-16-I and SO5-I seem have residual HEG gene nucleotides (non-founctional nucleotide sequences). The reason why

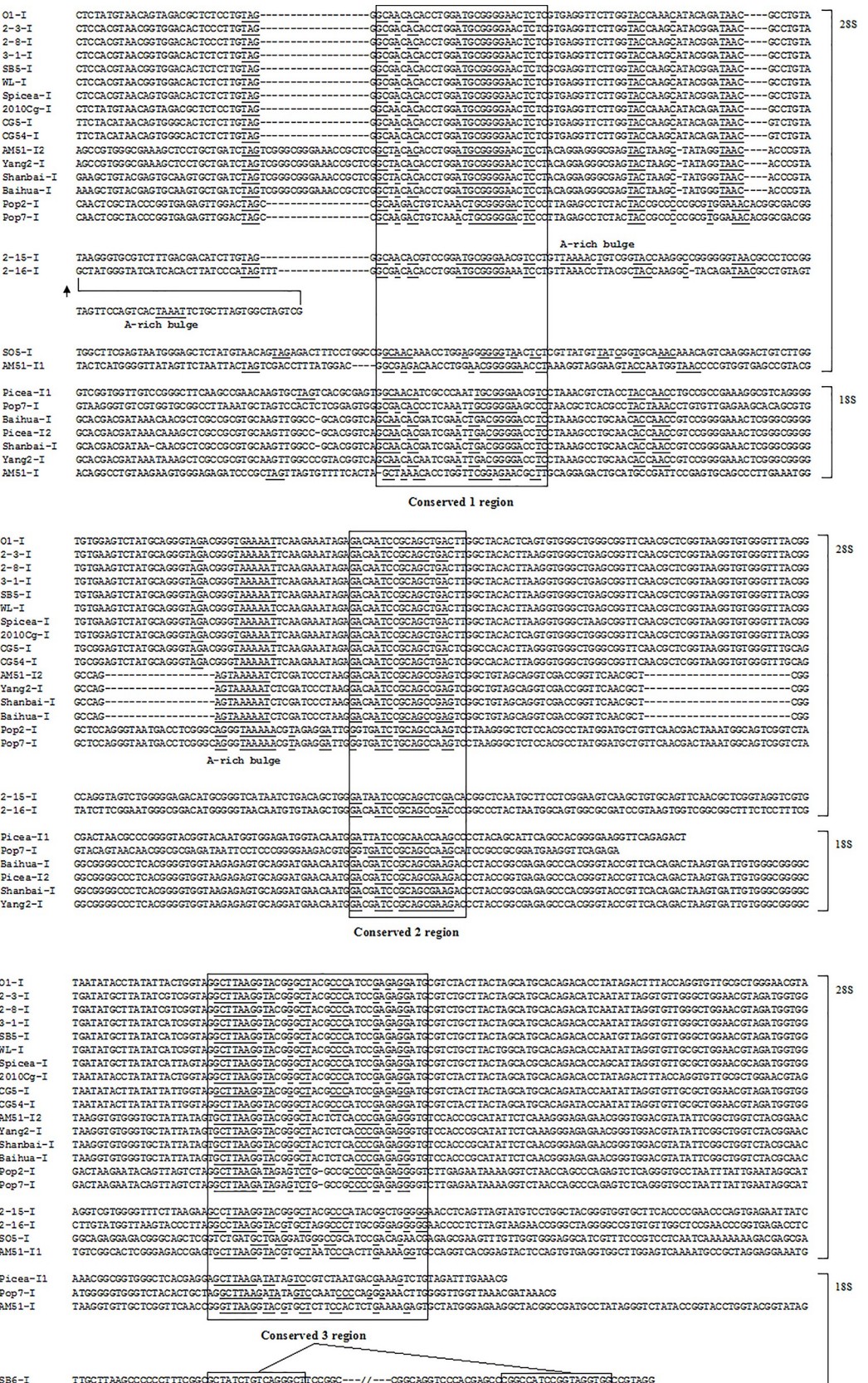

**Fig 5. Positions of conserved 1, conserved 2, and conserved 3 regions in 28S and 18S rDNA introns.** Top column of intron sequences are continuous from beginning to end. Below columns of intron sequences are extracted corresponding sequences. Identical nucleotides are underlined. Conserved 1, conserved 2, and conserved 3 regions are originally found in 28S rDNA introns.

residual HEG gene (non-founctional nucleotide sequences) still remain in intron sequences, probably because residual HEG genes have nucleotides which take part in intron secondary structure maintaining or founctions. We did not find the introns containing full length HEG gens, three introns containing HEG (SB5-I from 18S rDNA, SO5-I and 2-16-I from 28S rDNA) all contain residual HEG genes about 100–200 nucleotide sequences, from our isolated ectomycorrhizal fungal samples, our sample all were collected China.

The 12 nt nucleotide sequences TACCACAGGGAT at site 2 in the 3'-end of 28S rDNA, which is just upstream or downstream of the intron insertion position, the high conserved regions and identical nucleotide sequences in the introns at site 2, maybe much easier for introns to insert or delete. Introns break the integrality of exons sequences, introns possibly could control exon genes expressing. we can find 18S rDNA and 28S rDNA absence and presence of introns in the same isolate, for example, isolate CG5 has both 18S rDNA absence and presence of introns. We also find other isolates have both 18S rDNA absence and presence of introns. Genome DNA contains many 18S-5.8S-28S rDNA repeat unit, if product protein expressing from 28S rDNA is over-expressed more than cell metabolization need, will accumlate in cell. Product protein expressing from 28S rDNA is larger than from 18S rDNA, over-expression of 28S rDNA probably increase the cells more burden than over-expression of 18S rDNA. So the mechanism of 28S rDNA expressing control maybe more convenient than 18S rDNA expressing control, intron maybe one of the gene expressing controls. The majority of isolates contain 18S and 28S rDNA introns from our population genetic structure analysis previously, which means isolates containing 18S and 28S rDNA introns are more popular than

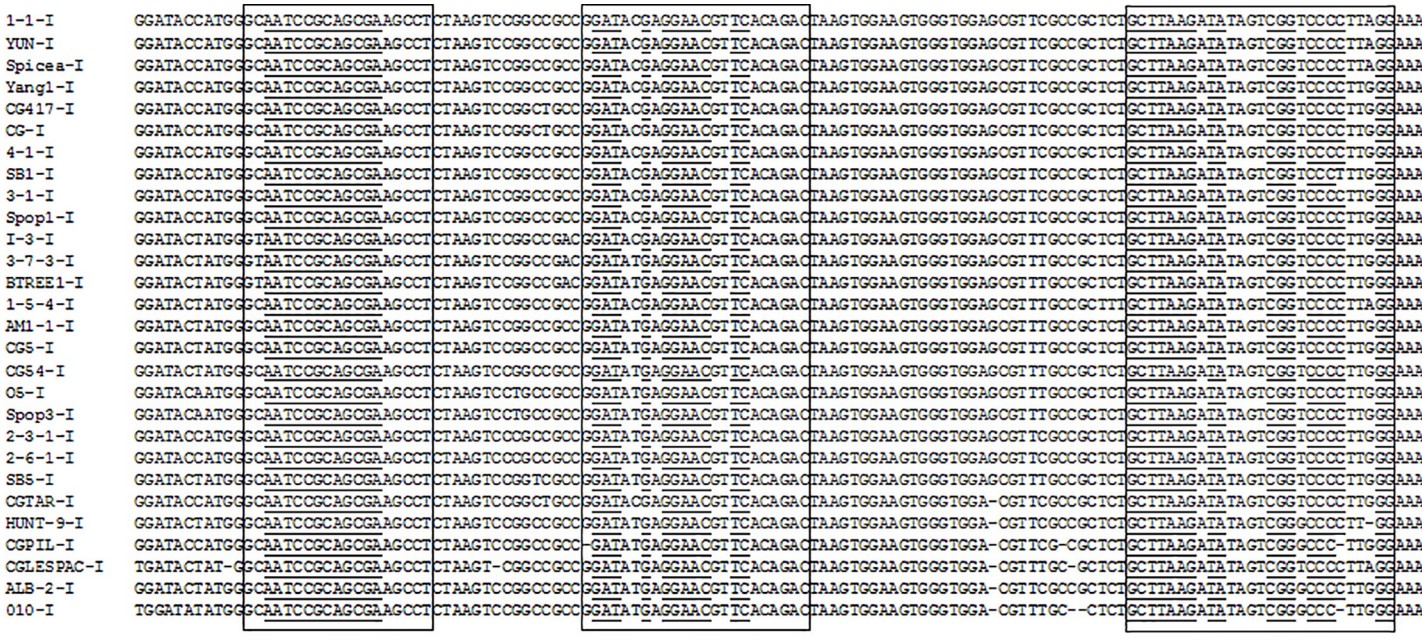

**Fig 6. Positions of conserved 1, conserved 2, and conserved 3 regions in *Cenococcum geophilums* 18S rDNA introns.** Intron sequences are continuous from beginning to end. Identical nucleotides are underlined. Conserved 1, conserved 2, and conserved 3 regions are originally found in 28S rDNA introns.

isolates without 18S and 28S rDNA introns, furthermore, which imply that isolates containing 18S and 28S rDNA introns fit selection pressure better than isolates without 18S and 28S rDNA introns. Probably, the population genetic structure with absence and presence of 18S and 28S rDNA introns are in the balance of gain and lost 18S and 28S rDNA introns. The presence rate of *Cenococcum geophilums* 18S rDNA introns from China, America, Europe is significantly different from reports and our work, maybe the presence rate of 18S rDNA introns fit the selection pressure coming from its geographical origin. Europe temperature overall is colder than China, whether the presence rate of introns and evolution speed of plant host and fungus are affected by temperature?

Weeks and Cech reported that the yeast mitochondrial group I intron b15 undergoes self-splicing at high $Mg^{2+}$ concentrations, but requires the splicing factor CBP2 for reaction under physiological conditions. Protein CBP2 could help assembly of the catalytic core, which involves association of two domains with each other and with other peripheral structures, and help association of the 5' domain containing the 5' splice site with the catalytic core properly [43]. The *Tetrahymena* preribosomal RNA intron could undergoes self-splicing in the absence of any proteins [44, 45]. Analysis the P1-IGS-P10 tertiary helix between 5'-end introns and exons in 18S and 28S rDNA in this study, we found that the complementary base pairing around the splicing sites were weak. In the P1-IGS-P10 tertiary helix around the splicing sites, there are many UG base pairing and unpairing bases. One of the group-I intron features known to be conserved is the last exon base U. UA and UG bonds are weaker than CG bond, and the presence of unpairing bases could also make the complementary base pairing helix unstable in same degree. The 5' and 3' exons both base pair to the intron's IGS resulting in P1 and P10 helix formation, respectively [45], UG base pairing and unpairing bases in P1-IGS-P10 tertiary helix between 5'-end introns and exons maybe make introns easy to be cut off and make 5' and 3' exons easy to be ligation. Other papers indicated that 5' splice site in P1-IGS-P10 tertiary helix possess UG bond quite common, in almost all introns present a UG pair at the 5' splice site [24, 46–49].

From the results above, introns in 28S rDNA are much easier to find conserved 1, 2, 3 region than introns in 18S rDNA; site 3 in 18S rDNA introns and site 2 in 28S rDNA introns are hot positions for intron insertion, introns located at site 3 in 18S rDNA and site 2 in 28S rDNA are much easier to find conserved 1, 2, 3 regions than site 1, 2, 4 in 18S rDNA introns and site 1 in 28S rDNA introns; *Cenococcum geophilums* is one of the most popular ectomycorrhizal fungi, introns in both 18S rDNA and 28S rDNA are much easier to find conserved 1, 2, 3 regions than other fungal species. It seems that the more convenient host sites, intron sequences and secondary structures, or isolates for 18S and 28S rDNA intron insertion and deletion, the more popular they are.

## Supporting information

**S1 File.**
(DOC)

## Acknowledgments

We thank professor Zhiwu Li from School of Mechano-Electronic Engineering of XiDian University of China for assistance in data analysis.

## Author Contributions

**Data curation:** Yu Wang.

**Formal analysis:** Ting Wang.

**Methodology:** Jian Liu, Zhuo Yu.

**Resources:** Wei Yan.

**Supervision:** Li-hong Chen.

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
