## [Decision Letter · Decision Letter 0]

21 Sep 2020

PONE-D-20-22370

Analysis of small and large subunit rDNA introns from several ectomycorrhizal fungi species

PLOS ONE

Dear Dr. Chen,

Thank you for submitting your manuscript to PLOS ONE. After careful consideration, we feel that it has merit but does not fully meet PLOS ONE’s publication criteria as it currently stands. Therefore, we invite you to submit a revised version of the manuscript that addresses the points raised during the review process.

We look forward to receiving your revised manuscript.

Kind regards,

Erika Kothe

Academic Editor

PLOS ONE

Journal Requirements:

2. In your Methods section, please provide additional location information, including geographic coordinates for the data set if available.

'We thank professor Zhiwu Li from School of Mechano-Electronic Engineering of XiDian University of China for assistance in data analysis.'

a. Please complete your Competing Interests statement to state any Competing Interests. If you have no competing interests, please state "The authors have declared that no competing interests exist.", as detailed online in our guide for authors at http://journals.plos.org/plosone/s/submit-now

5. Please amend the manuscript submission data (via Edit Submission) to include authors Mark Gomelsky and Larry R Peterson.

Additional Editor Comments:

The reviewer gave very good suggestions of how to improve the submission. In general, it must be made clear that the analyzed strains were pure, were what was identified when looking at the rhizomorphs, and also to compare to genomic data to make sure that different rRNA genes present in one strain (multiple rRNA genes in one genome as well as two nuclei in the respective dikaryon that was isolated) are not mixed.

Reviewers' comments:

Reviewer's Responses to Questions

**Comments to the Author**

1. Is the manuscript technically sound, and do the data support the conclusions?

Reviewer #1: Partly

2. Has the statistical analysis been performed appropriately and rigorously? 

Reviewer #1: N/A

3. Have the authors made all data underlying the findings in their manuscript fully available?

Reviewer #1: Yes

4. Is the manuscript presented in an intelligible fashion and written in standard English?

Reviewer #1: No

5. Review Comments to the Author

Reviewer #1: The manuscript by Chen et al. analyzes introns from small and large subunit ribosomal DNA (rDNA) from several ectomycorrhizal fungal species. Specifically, the authors studied the intron insertion position, intron sequence homology, secondary structure comparison, and tried to shed light on the evolutionary origin between 18S and 28S rDNA introns in those fungal species. However, this study still seems to be ‘half-baked’ and warrant further investigation/analysis.

It would be suitable for publication if the following points are addressed (major revisions).

1.Agarose gel images of the PCR amplicons could be a good additive in the text. May be include as supplementary figs. The gel images would give a quick view on the presence and absence of introns based on the size variations. What were the size of the amplicons? How clean was the PCR amplification?

2.The authors indicated that ‘Pure cultures of tested strains were isolated from sclerotial bodies as well as mycorrhizae samples which were collected….’ How did they ensure ‘purity’ in the metacommunity at the time of sampling? If these are already existing stock cultures, please provide ATCC or MTCC accession numbers in the table. This is important to know as any minute ‘noise/impurity’ would be amplified through PCR/sequencing, which would further affect the downstream analysis.

3.Ancestral state reconstructions/evolutionary framework could be achieved by inferring gains and losses of introns genome‐wide and not only within rDNA regions, because introns with HEG(s) have potential to invade other genomic regions with significant sequence similarities. Please highlight the number of times complete or partial intron loss has occurred across the evolutionary history of the studied fungal species. Overall, the evolutionary origin between 18S and 28S rDNA introns is unclear from this study.

4.For introns containing HEGs, it would be desirable to check whether the homing endonucleases (HEs) are active or not. It would be interesting to see if the reconstructed/cloned HE ORF overexpressed protein cuts the homing site, which could further support this study.

In general, the manuscript suffers from serious grammatical errors, fewer spelling mistakes, along with font inconsistencies and poorly structured sentences that needs thorough attention before publication.

Decision: MAJOR REVISIONS requested.

Tuhin K. Guha, Ph.D.

6. PLOS authors have the option to publish the peer review history of their article (what does this mean?). If published, this will include your full peer review and any attached files.

Reviewer #1: **Yes: **Tuhin K. Guha, Ph.D.

---

## [Author Response · Author response to Decision Letter 0]

3 Jan 2021

I agree with reviewer and editor comments.

---

## [Decision Letter · Decision Letter 1]

7 Jan 2021

Analysis of small and large subunit rDNA introns from several ectomycorrhizal fungi species

PONE-D-20-22370R1

Dear Dr. Chen,

We’re pleased to inform you that your manuscript has been judged scientifically suitable for publication and will be formally accepted for publication once it meets all outstanding technical requirements.

Kind regards,

Erika Kothe

Academic Editor

PLOS ONE

Additional Editor Comments (optional):

Reviewers' comments:

Reviewer's Responses to Questions

**Comments to the Author**

1. If the authors have adequately addressed your comments raised in a previous round of review and you feel that this manuscript is now acceptable for publication, you may indicate that here to bypass the “Comments to the Author” section, enter your conflict of interest statement in the “Confidential to Editor” section, and submit your "Accept" recommendation.

Reviewer #1: All comments have been addressed

2. Is the manuscript technically sound, and do the data support the conclusions?

Reviewer #1: Yes

3. Has the statistical analysis been performed appropriately and rigorously? 

Reviewer #1: N/A

4. Have the authors made all data underlying the findings in their manuscript fully available?

Reviewer #1: Yes

5. Is the manuscript presented in an intelligible fashion and written in standard English?

Reviewer #1: Yes

6. Review Comments to the Author

Reviewer #1: (No Response)

7. PLOS authors have the option to publish the peer review history of their article (what does this mean?). If published, this will include your full peer review and any attached files.

Reviewer #1: **Yes: **Tuhin Guha

---

## [Editor Report · Acceptance letter]

3 Mar 2021

PONE-D-20-22370R1 

Analysis of small and large subunit rDNA introns from several ectomycorrhizal fungi species 

Dear Dr. Chen:

I'm pleased to inform you that your manuscript has been deemed suitable for publication in PLOS ONE. Congratulations! Your manuscript is now with our production department. 

Kind regards, 

on behalf of

Prof. Dr. Erika Kothe 

Academic Editor

PLOS ONE